# Personalized Nutritional Strategies to Reduce Knee Osteoarthritis Severity and Ameliorate Sarcopenic Obesity Indices: A Practical Guide in an Orthopedic Setting

**DOI:** 10.3390/nu15143085

**Published:** 2023-07-09

**Authors:** Hassan Zmerly, Marwan El Ghoch, Leila Itani, Dima Kreidieh, Volkan Yumuk, Massimo Pellegrini

**Affiliations:** 1Orthopedics and Traumatology Unit, Villa Erbosa Hospital, 40129 Bologna, Italy; zmerly@msn.com; 2Ludes Campus, 6912 Lugano, Switzerland; 3Department of Nutrition and Dietetics, Faculty of Health Sciences, Beirut Arab University, Riad El Solh, Beirut P.O. Box 11-5020, Lebanon; l.itani@bau.edu.lb (L.I.); d.kraydeyeh@bau.edu.lb (D.K.); 4Faculty of Medicine, UniCamillus—Saint Camillus International University of Health and Medical Sciences, Via di Sant’Alessandro, 8, 00131 Rome, Italy; 5Department of Biomedical, Metabolic and Neural Sciences, University of Modena and Reggio Emilia, Via Giuseppe Campi, 287, 41125 Modena, Italy; massimop@unimore.it; 6Division of Endocrinology, Metabolism and Diabetes, Istanbul University Cerrahpaşa Medical Faculty, Istanbul 34452, Türkiye; vdyumuk@istanbul.edu.tr

**Keywords:** body composition, total body fat, fat distribution, body lean mass, muscle mass, muscle strength, obesity, osteoarthritis, sarcopenic obesity, skeletal muscle mass

## Abstract

Knee osteoarthritis (KOA) is one of the most common joint diseases, especially in individuals with obesity. Another condition within this population, and which presents frequently, is sarcopenic obesity (SO), defined as an increase in body fat and a decrease in muscle mass and strength. The current paper aims to describe recent nutritional strategies which can generally improve KOA clinical severity and, at the same time, ameliorate SO indices. Searches were carried out in the PubMed and Science Direct databases and data were summarized using a narrative approach. Certain key findings have been revealed. Firstly, the screening and identification of SO in patients with KOA is important, and to this end, simple physical performance tests and anthropometric measures are available in the literature. Secondly, adherence to a Mediterranean diet and the achievement of significant body weight loss by means of low-calorie diets (LCDs) remain the cornerstone nutritional treatment in this population. Thirdly, supplementation with certain micronutrients such as vitamin D, essential and non-essential amino acids, as well as whey protein, also appear to be beneficial. In conclusion, in the current review, we presented a detailed flowchart of three different nutritional tracks that can be adopted to improve both KOA and SO based on joint disease clinical severity.

## 1. Introduction

Osteoarthritis (OA) is a chronic degenerative joint disease, characterized by progressive deterioration and can involve the articular cartilage, the synovium, the ligaments, as well as the subchondral bone [1]. The prevalence of symptomatic OA is estimated in nearly 10–20% of those over 60 within the general population [2] and is continuously increasing [3]. The symptoms may vary, based on the clinical severity and the time of onset of OA, which can be completely asymptomatic but can progress to pain, stiffness and physical limitation, reaching complete movement invalidity in certain cases and mainly affecting the vertebrae, the knees and the hip joints due to their role in sustaining body weight [4].

The etiopathogenesis is not fully clarified; however, the most acceptable hypothesis is a damage-repairing impairment of cartilage tissue due to an interaction of mechanical (i.e., body weight, postural problems), biochemical and genetic factors [5]. In particular, KOA is a common progressive multifactorial joint disease, considered one of the major causes of disability among the elderly [6]. In 2020, the global prevalence of KOA was estimated to be roughly 25% in middle aged and older adults (i.e., over 40 years), and could lead to significant and unavoidable impacts on health, workplace productivity and economic costs [7].

Certain factors have been identified which could increase the risk of KOA, some of which are non-modifiable, such as age and sex (i.e., being a female), but others are modifiable (i.e., injuries, body weight status, education, etc.) [8].

Therefore, it is crucial to identify the modifiable risk factors by providing effective preventive strategies in the early stage of the disease (i.e., KOA) to ameliorate it or to arrest/reduce its progression to advanced stages. Obesity was shown to be the most significant modifiable risk factor for the development and progression of KOA [9], whereby individuals with obesity (BMI ≥ 30.0 kg/m^2^) are seven times more likely to develop KOA than their normal-weight counterparts (BMI < 25.0 kg/m^2^) [10]. Moreover, recent literature considers sarcopenia as a potential risk factor for KOA; however, the available literature is still limited [11].

In this context, over the last few decades, there has been a growing interest in a new phenotype that combines obesity and sarcopenia; it is termed sarcopenic obesity (SO) and results in a reduction in muscle mass and strength and an increase in body fat deposition [12]. On a general scale, several investigations have shown that patients with SO have a higher prevalence of obesity-related comorbidities (i.e., type 2 diabetes, hypertension, dyslipidemia, etc.). With respect to non-sarcopenic obesity [13,14,15], specific, recent literature has reported the negative impact of SO on KOA [16] and recommends routine screening for this condition during the initial assessment so as to better manage its treatment [16].

Based on these considerations, the purpose of the current narrative review is to summarize the recent available literature (i) to identify simple screening and the identification tools of SO in patients with KOA and (ii) to propose a nutritional protocol with specific strategies that can primarily improve the clinical severity of KOA and simultaneously ameliorate SO. Therefore, such a report may have relevant clinical implications, especially in an orthopedic setting.

## 2. Materials and Methods

This current review has been prepared in adherence to the narrative review guidelines adopted by the Academy of Nutrition and Dietetics (Appendix A) [17]. We searched the PubMed and Science Direct databases for relevant literature related to KOA and SO using the medical subject headings (MeSH) and their combinations: #1 = sarcopenic obesity, #2 = lean mass, #3 = muscle mass, #4 = muscle strength, #5 = physical performance, #6 = knee, #7 = osteoarthritis, #8 = cartilage damage, #9 = WOMAC, #10 = knee osteoarthritis, #11 = pain, #12 = stiffness and #13 = clinical outcome. An additional manual search was carried out to retrieve the other scientific literature that had not been identified via the initial strategy.

All analyses that evaluated KOA and SO were considered and included in the narrative review if they were (i) manuscripts in English, (ii) original articles and were (iii) prospective or retrospective observational (analytical or descriptive), experimental or quasi-experimental studies. Non-original studies, including editorials and letters to the editor, were excluded. No limitations on the date of publications were imposed; however, the focus was on recent literature (i.e., the last five years) in order to report updated findings.

## 3. Results

### 3.1. Diagnosis of KOA

After considering the medical history and following a physical examination, X-ray and/or ultrasound, the diagnosis of KOA is confirmed or rejected [18] and is reported using the Kellgren and Lawrence classification system [19]: Grade 0: no radiological findings of OA; Grade I (doubtful): doubtful joint space narrowing and possible osteophytic lipping; Grade II (minimal): certain osteophytes and possible joint space narrowing; Grade III (moderate): moderate multiple osteophytes, a certain narrowing of the joint space, some sclerosis and possible deformity of bone ends; Grade IV (severe): large osteophytes, marked narrowing of joint space, severe sclerosis and certain deformity of bone ends (Table 1; Figure 1).

### 3.2. Screening/Diagnosis of SO in Patients with KOA

Few performance tests are available to screen SO among the KOA population specifically; in fact, to the best of our knowledge, Godziuk and colleagues recently found that grip strength adjusted by BMI, with cut-points less than 0.65 in females and 1.1 in males, is indicative of a patient with KOA being at a higher risk of having SO, and can be considered a simple screening tool [20] (Table 1). Once identified, a more accurate assessment should be sought to confirm or reject the diagnosis of SO and, to this aim, appendicular lean mass (ALM) adjusted by body size, including body weight (kg) or BMI (kg/m^2^), seems to be clinically reasonable in the population in question. Specifically, in specialized settings related to KOA (i.e., rehabilitation medicine), it was found that ALM/BMI cut-points are more sensitive than other indices (i.e., ASM/height^2^ cut-points) in identifying SO in a clinical population with KOA. These cut-points of ALM/BMI are <0.512 in females and <0.789 in males [20] (Table 1).

Therefore, the impaired (hand grip strength)/BMI combined with ALM/BMI appears to be a suitable method of confirming the diagnosis of SO in patients with KOA, since it assesses the reduction in both dimensions of the SO definition (Table 1).

### 3.3. Nutritional Management in Patients with KOA and SO Diets

(1)Mediterranean Diet

The Mediterranean diet (MD) eating pattern is characterized by a high consumption of plant foods (i.e., whole grains, fruits, vegetables, legumes, nuts, seeds and olives), a moderate consumption of dairy products (i.e., mainly cheese and low-fat yogurt), no more than four eggs per week, fish at least twice a week, a low intake of sweets and red meat and a moderate intake of alcohol (i.e., mostly red wine during meals), with the main source of fat being extra virgin olive oil [21]. Several studies have demonstrated the beneficial impact of the MD on health, and the potential mechanisms behind the pro-health effects of the MD seem to be mainly due to the anti-inflammatory properties of this diet [22], that derive from different nutrients and foods that interact between each other reinforcing their beneficial effects [23]. Therefore, one of the main protagonists is extra virgin olive oil, rich in polyphenols and known for its antioxidant and anti-inflammatory properties [24].

In addition, the MD is also rich in fibers, which seem to act on the intestinal microbiota by modulating its composition, activity and the production of metabolites that help regulate the inflammatory pathways [25]. More specifically, the high fiber diet increases certain intestinal bacterial species that determine the production of short-chain fatty acids such as acetate and butyrate [26,27], important for the prevention of inflammatory diseases [27]. For instance, butyrate can reduce the production of pro-inflammatory molecules such as TNF-a, IL-1b and nitric oxide; reduce the activity of the nuclear factor kappa-light-chain-enhancer of activated B cells (NF-OEB); inhibit the production of IL-12 and increase the production of IL-10 by monocytes [28,29].

Finally, the MD is rich in omega-3 PUFA from fish and vegetable sources and has an adequate omega-6/omega-3 ratio [30], thereby promoting a better inflammatory profile compared to other Western pattern diets. The intake of omega-6 fatty acids in the case of the latter is greater, which induces a higher production of pro-inflammatory cytokines that increase the risk of chronic diseases such as type 2 diabetes and atherosclerosis [31]. In fact, the dietary omega-3 fatty acids are characterized by a variety of anti-inflammatory effects and seem to be able to reduce the inflammatory process [32].

In the context of this review, in large community dwellings, a cross-sectional study revealed that a greater adherence to the MD is associated with a lower prevalence of KOA [33]. However, among the components of the MD, only a higher consumption of cereals was associated with lower odds of having KOA [33]. Another cross-sectional study, conducted in 2018 on patients affected by KOA, showed that a higher adherence to the MD in this population is associated with better knee cartilage morphology in terms of volume and thickness, assessed using magnetic resonance (MRI) [34]. In 2019, Veronese and colleagues published a four-year longitudinal study on the impact of the MD on KOA; they found that a greater adherence to the MD was related to a lower risk of pain worsening, or the onset of symptoms of non-symptomatic forms of KOA [35] (Figure 2).

Despite controversial opinions regarding the adherence to the MD and sarcopenia, a recent cross-sectional study among a large sample of community-dwelling, older adults [36] showed that lower adherence scores to the MD were associated with lower handgrip strength; the authors speculate that these individuals are at a higher risk of developing sarcopenia [36]. On the other hand, a recent review which focused on extra virgin olive oil (central component of the MD) and its potential benefits in maintaining skeletal muscle homeostasis by increasing skeletal muscle protein synthesis rates and stimulating an anabolic muscle response highlighted an attenuation in muscle wasting and a delay in sarcopenia progression [37]. These benefits of extra virgin olive oil seem to be related to the polyphenols, which demonstrate a clear ability to activate anabolic pathways and to counteract age-/disease-related changes involved in muscle degeneration, such as mitochondrial alterations [38]. In particular, several studies underlined the role of extra virgin olive oil in maintaining mitochondrial homeostasis, in light of the fact that accumulation of dysfunctional mitochondria is a major contributing factor to the development of sarcopenia [39,40]. However, these findings need to be interpreted with caution (Figure 3).

(2)Weight loss

In the case of the presence of excess body weight (i.e., obesity), weight loss (WL) is necessary and is highly recommended to reduce pain and stiffness and to improve knee joint function [41]. The extensive literature available on intentional WL among populations with obesity and KOA has mainly been included in three important systematic reviews and meta-analyses conducted over the last 15 years [42]. The first, carried out in 2007 by Christensen and colleagues, suggested that patients with obesity and KOA should lose at least 7.5% of their weight in order to obtain a moderate improvement in self-reported disability [42]. A more recent systematic review and meta-analysis, conducted in 2018 by Chu and colleagues, showed that a WL% of between 5 and 10% was needed to significantly improve pain and self-reported disability, and to improve quality of life (QoL) [43]. Finally, a large network meta-analysis, conducted in 2020 by Panunzi and colleagues, which relied on the Western Ontario and McMaster Universities’ (WOMAC) OA index to identify the KOA clinical severity, found an improvement in pain, function and stiffness scores by 2% points for every unit of WL% [44]. Interestingly, the authors conducting this network analysis also compared different WL interventions (i.e., bariatric surgery, low-calorie diet (LCD), very-low-calorie diet (VLCD), cognitive behavioral therapy for obesity (CBT-OB), etc.) with the improvement of KOA symptoms and clinical conditions in patients with obesity, and found that an LCD, when combined with exercise, was the most effective dietary intervention in terms of reducing pain in this population, followed by a VLCD [44] (Figure 2).

Weight loss also appears to ameliorate SO in patients with obesity, as demonstrated in a recent longitudinal study [45]. Specifically, a Mediterranean LCD (1200–1500 Kcal), combined with an active lifestyle (8000–10,000 steps/day) in a weight management setting with a WL of at least 5% determined an improvement in the appendicular skeletal-mass-to-weight ratio, indicating a decrease in the prevalence of SO by 12.2% [45]. Moreover, this finding was confirmed by logistic regression analysis, revealing a significant WL% ≥ 5%, which decreased the risk of SO by 91% after adjusting for age and gender [45] (Figure 2).

(3)Very-Low-Calorie Ketogenic Diet (VLCKD)

A VLCKD is characterized by a restriction of carbohydrates and total daily energy intake and an increase in fat and protein, proposed as a highly effective dietary strategy for patients who require rapid WL in the short term (i.e., before surgeries) in order to reduce the level of obesity and its associated risk factors [46]. It is important to highlight that WL achieved with VLCKD is predominantly body fat mass, while muscle mass loss is limited [47]. Recently a VLCKD protocol for obesity management in adults has been proposed by the European Association for the Study of Obesity (EASO) which suggests <800 Kcal/day, with a carbohydrate restriction of 30–50 g/day (≃13% of total energy intake), an increase of 30–40 g/day (≃44%) of fats and approximately 0.8–1.2 g/day of proteins per kilogram of body weight (≃43%) [47]. This protocol requires strict guidance by a team of qualified specialists and careful monitoring. It is divided into three phases: active (or ketogenic), re-education and maintenance/transition to a MD [47].

In this scenario, in general, an increase in ketone bodies in the blood after ketogenic diets (i.e., VLCKD), such as β-hydroxybutyrate (βOHB), seem to act as a signal metabolite that can suppress certain inflammatory pathways [48]. Specifically, this is the nucleotide-binding domain, leucine-rich repeat, and pyrin domain-containing protein 3 (NLRP3) inflammasome, where its activation seems to promote the secretion of proinflammatory cytokines [49]. An increasing number of literature studies have reported that the NLRP3 protein is significantly upregulated in OA cartilage by increasing the secretion of interleukin-1 β (IL-1β) and IL-18 [50] (Figure 3).

Therefore, NLRP3 inflammasome down regulation was suggested as a promising strategy for OA treatment by inhibiting excessive inflammatory responses [51,52]. In fact, a recent study, conducted in an animal model, investigated the effect of a ketogenic diet (vs. a standard diet) on OA in rats [53]. The investigators of this study found that a ketogenic diet can protect the articular cartilage and subchondral bone in a rat OA model by inhibiting NLRP3 inflammasome and reducing the OA inflammatory response (i.e., IL-1β and IL-18) [53] (Figure 3). This finding may provide a new direction for future research in relation to the treatment of OA in humans since a ketogenic diet can reduce inflammation by inhibiting the NLRP3 inflammasome in patients with KOA (Table 2; Figure 2).

Similarly, there are few emerging findings regarding the impact of VLCKD on SO [54,55]; only a recent pilot study conducted in humans investigated the effect of VLCKD on body composition and physical performance in patients with SO [55]. The authors found that VLCKD, combined with physical exercise training, reduces adipose depots and preserves fat free mass with a preservation of muscle strength during WL [55] (Table 2; Figure 2). This finding, if confirmed in a larger population, could represent a useful strategy for ameliorating SO.

### 3.4. Supplementation

(1)Vitamin D

A recent systematic review has shown that vitamin D deficiency is associated with a higher risk of KOA progression [56], and recently this has been reported to be potentially linked to SO [57]. In this regard, certain authors have speculated that vitamin D supplementation may have benefits in terms of musculoskeletal health [58].

In 2021, a systematic review and meta-analysis was published with the aim of providing evidence on the effects of vitamin D supplementation as a strategy for the prevention and clinical treatment of KOA [59]. The results of the meta-analysis conducted on more than 1500 patients with KAO showed that vitamin D supplementation significantly improves the total WOMAC score in patients with KOA, as well as the WOMAC pain, function and stiffness subscales. The investigators conducting this meta-analysis concluded that vitamin D supplementation improves the function and symptoms (i.e., pain and stiffness) in patients with KOA, but it appears unable prevent the structural progression of the disease (i.e., KOA) [59]. Specifically, according to a subgroup analysis, an improvement in stiffness is achievable with less than 2000 IU/day vitamin D supplementation [59] (Table 2; Figure 2). This is of particular interest since overdose of vitamin D supplementation is not always beneficial but, on the contrary, it can be harmful; for instance, in a recent randomized clinical trial on healthy adults, supplementation with vitamin D for three years at a dose of 4000 IU per day or 10,000 IU per day, compared to lower doses (i.e., 400 IU per day), resulted in significantly lower bone mineral density (i.e., radial BMD) [60], increasing the risk of fractures [61].

On the other hand, recent well-designed studies assessed the effects of a six-month vitamin D supplementation (10,000 IU three times a week) on SO indices and reported a major improvement in the appendicular lean mass (ALM), measured by dual-energy X-ray absorptiometry (DEXA), with respect to the placebo group [62] (Figure 2). However, some studies were not able to confirm the beneficial impact of vitamin D supplementation on SO [63]; therefore, further investigations are required in this regard.

(2)Antioxidant Supplements

The reactive oxygen species (ROS) are instable and are highly reactive molecules, usually involved in several physiological functions (i.e., the modulation of cell survival, cell death, differentiation, cell signaling, etc.) [64]. However, under certain circumstances, environmental stressors (e.g., ionizing radiations, pollutants and heavy metals) produce an increase in ROS production/accumulation and a decrease in their neutralization through the antioxidant system; this imbalanced phenomenon is defined as oxidative stress [65]. A wide range of studies have demonstrated that OS contributes to the onset and progression of several clinical conditions, i.e., diabetes, cardiovascular and neurodegenerative diseases as well as cancers [66].

Recently, oxidative stress has been suggested as playing a role in the physiopathology of OA and SO [67,68]. For this reason, the use of antioxidant supplementation has been proposed as a potential strategy to prevent and slow the progression and treat both clinical conditions (i.e., KOA and SO) [69].

Several studies have reported the benefits of antioxidant supplementation for pain management and function in patients with KOA that were based on curcumin, avocado, Boswellia and soya bean, etc.; however, other studies have demonstrated the opposite [70,71]. For this reason, a recent systematic review and meta-analysis assessed the impact of antioxidants on KOA symptoms and found that the overall effect of antioxidant supplementation had no significant effect on the total, difficulty, pain and stiffness WOMAC index scores, showing little success in the treatment and relief of the clinical symptoms of KOA [72].

On the other hand, the onset and progression of SO are related to oxidative stress, inflammation and insulin resistance. Recent evidence suggests that flavonoids as antioxidants may be effective in preventing and treating SO due to their ability to improve insulin sensitivity and reduce oxidative stress, inflammation, and mitochondrial dysfunction [69]. However, very few clinical trials in humans have investigated the effectiveness of flavonoids in SO. Therefore, further studies are still needed to assess the potential impact of flavonoids in the prevention/treatment of SO [69].

(3)Amino Acid Supplementation

Amino acids are the basic building units of proteins, and there are 20 common amino acids found in proteins [73]. They can be classified into two categories: (i) non-essential amino acids (non-EAAs), of which there are 11, usually produced in the body, including alanine, arginine, asparagine, aspartic acid, cysteine, glutamic acid, glutamine, glycine, proline, serine and tyrosine [74] and (ii) essential amino acids (EAAs), which the body is unable to produce and should, therefore, be introduced through the diet. There are nine EAAs: phenylalanine, threonine, tryptophan, methionine, lysine, histidine, valine, leucine and isoleucine [75]. The last three are known as essential branched-chain amino acids, involved in muscle metabolism by stimulating growth, re-generation and repair [76].

Certain early studies found that the ratio of serum branched-chain amino acids to histidine may have potential clinical use as a KOA biomarker [77]; very recently, certain studies revealed that the serum levels of leucine and glutamic acid are potential biomarkers and are associated with a higher sarcopenic risk in individuals with type 2 diabetes [78].

EAA Supplementation, KOA and SO

Recent data deriving from a recent systematic review revealed the efficacy of EAA supplementation, where reasonable evidence is available regarding the beneficial impact of EAA supplementation during orthopedic peri- and post-surgical settings of KOA management [79]. For instance, a double-blind, placebo-controlled RCT investigated the effect of a twice-daily ingestion of 20 g of EAAs between meals for one week before and two weeks after total knee arthroplasty (TKA) [80]. Patients who received EAAs exhibited less quadriceps muscle atrophy between the baseline and two-to-six weeks after surgery compared to the placebo group, which experienced greater atrophy [80]. Moreover, the EAA supplementation also seems to attenuate atrophy in the non-operated quadriceps and in the hamstring and adductor muscles of both extremities [80]. Finally, the patients who took EAAs performed better in terms of functional motility at two and six weeks after surgery, and the authors claimed that EAA treatment attenuated muscle atrophy and accelerated the return of functional mobility in adults following TKA [80]. In 2018, the same research group carried out a similar study but with the unique difference of extending the supplementation to six weeks after surgery and confirmed their previous findings; in addition, they were able to guarantee the safety of this supplementation [81] (Table 2; Figure 2).

On the other hand, in terms of SO, recent findings have reported that SO is associated with lower serum levels of branched-chain amino acids with respect to patients with obesity only but not sarcopenia [82]. Moreover, just recently, Zhou and colleagues investigated the effect of EAA supplementation when treating SO in male adults aged over 60 [83]. The oral EAA mixture was administered twice a day with a total of 10 g/day (leucine 1.7 g; lysine 1.3 g; isoleucine 1.3 g; valine 1.4 g; threonine 1.0 g; phenylalanine 2.0 g; methionine 1.0 g and tryptophan 0.3 g) for a period of 28 weeks, and follow-up assessments were completed at 12, 20 and 28 weeks; an improvement in SO was noticed only after 28 weeks [83] (Table 2; Figure 2).

Non-EAA Supplementation and KOA

To the best of our knowledge, no study is available on the effect of non-EAA supplementation on KOA or SO. However, a recent randomized, double blind, placebo-controlled trial examined the effects of a combination of six non-EAAs (alanine, aspartic acid, glutamic acid, glycine, proline and serine) in an adult population with knee joint discomfort but with no clear diagnosis (i.e., KOA) in comparison with the placebo group [84]. The first group took 12 g of the non-EAA formulation orally (4 g, three times a day) for a period of 12 weeks, while the control group took equivalent doses of a placebo. Symptoms were evaluated through validated questionnaires at baseline, 4 and 12 weeks [84]. The non-EAA group reported significant improvements in joint pain, discomfort, and stiffness, both during the resting state and during normal activity after four weeks of supplementation [84]. This result is of particular interest with regard to the population with no evident KOA (i.e., Grade 0); however, they were referred to the orthopedic specialist for joint symptoms from prevention prescriptive (Table 2; Figure 2).

(4)Protein Supplementation

Whey protein supplements, KOA and SO

A recent study was conducted on females over the age of 60 with KOA (non-surgical population) who underwent 12 weeks resistance exercise training combined with whey protein supplementation or placebo in order to investigate whether the former displayed a major improvement in their sarcopenic indices (i.e., appendicular muscle mass, gait speed), level of physical activity and KOA clinical condition using the WOMAC index [85]. In this study, each serving supplementation of 24 g contained 66% of protein (whey 7.0 g, leucine 2.2 g), and all participants were asked to consume the protein supplement twice a day (one serving at breakfast and lunch, respectively, on the non-exercise day) [85]. On the training day, participants were instructed to consume two servings (one at breakfast and one within 30 min of resistance physical exercise) [85]. Compared to the placebo group, the individuals who took the protein supplement achieved an increase in the ALM index and gait speed, as well as levels of physical activity and the global WOMAC score [85]. Through a recent systematic review and meta-analysis of randomized trials, this finding has been confirmed among surgical populations with lower extremity OA, who underwent a joint replacement (i.e., knee and hip); the supplementation mainly contained whey proteins and was combined with exercise training for two-to-seven weeks of rehabilitation. Investigators noted improvements in muscle mass and strength, as well as pain, particularly among those who had undergone total joint replacement [86]. Another recent systematic review also confirmed the usefulness and safety of the whey protein supplementation during the peri-operative period of knee and hip replacement but, more interestingly, the researchers recorded limited evidence regarding the beneficial effects of creatine supplementation in this population [79] (Table 2; Figure 2).

On the other hand, a recent study conducted on animals demonstrated that whey protein supplementation, regardless of physical exercise, seems to have potential anti-obesity and anti-sarcopenic effects in the SO mice model [87]. In line with this finding, in an early study, whey protein appeared to promote the reduction in adipose tissue and increased muscle protein synthesis during caloric restriction-induced WL (≈7% from baseline) in elderly individuals with obesity [88]. Similarly, a recent randomized, double-blind, placebo-controlled trial, conducted in humans, investigated the effects of whey protein supplementation (vs. placebo) associated with physical exercise on body composition, muscular strength and functional capacity in females with SO [89]. The protocol of the supplementation was 35 g of hydrolyzed whey protein three times a week after each training session for 12 weeks, reporting a major improvement in the SO indices among the supplementation group with respect to the placebo group and, in particular, an increase in the ALM and a decrease in the total, trunk total and trunk fat mass [89] (Table 2; Figure 2).

**Table 2 nutrients-15-03085-t002:** Nutritional approach to management of individuals with KOA and SO and level of evidence.

	Grade	Nutritional Approach	Suggested Duration	Type of Evidence
Prevention program	0	Cereal-rich MD Non-EAAs	Lifelong	Cross-sectional study [33,34] Randomized controlled trials [84]
Lifestyle intervention (no surgery)	I and II	MD LCD (1200–1500 kcal/day) EAAs (10 g/day) Vit D (less than 2000 IU/day)	18 months	Observational study [35,45,85] Systematic review [42,43,59] Randomized controlled trials [62,83,89] Animal model study [87]
Intensive lifestyle intervention (surgery)	III and IV	Very-low-calorie diet (VLCD; 800 kcal/day) EAAs/whey protein (20 g/day) VLCKD (800 kcal/day)	5–7 weeks	Observational study [55] Systematic review [79] Randomized controlled trials [80,81] Animal model study [53] Case report [54]

Abbreviations: KOA = knee osteoarthritis; SO = sarcopenic obesity; MD = Mediterranean diet; ALM = appendicular lean mass; Vit D = vitamin D supplementation; EAA = essential amino acid supplementation; non-EAA = non-essential amino acid supplementation; LCD = low-calorie diet; VLCD = very-low-calorie diet; VLCKD = very-low-calorie ketogenic diet.

## 4. Discussion

The current narrative review represents an attempt to describe a kind of nutritional guide that can be used by health professionals (i.e., orthopedics, nutritionists, dieticians, obesity specialists, physical therapists, etc.) in the management of individuals with KOA, with obesity and with SO, the primary aim being to improve the clinical severity of KOA and at the same time ameliorate SO.

### 4.1. Interaction between KOA and SO

OA and sarcopenia are two key issues in public health and in many clinical settings, which usually co-exist among individuals with obesity; robust literature has identified the latter as a factor which increases the risk of having KOA, as well as SO [11]. Despite the fact that the underlying mechanisms behind this are still unclear and not fully understood, we speculate that obesity may act in a direct and indirect way on joints and muscles [90]. In fact, obesity is associated with sedentary behaviors [91] and may avoidably lead to muscle disuse (i.e., function) and/or atrophy (i.e., mass), facilitating the onset/deterioration of SO [92]. Similarly, excessive body weight can overload the joints and result in the onset/progression of KOA [93]. On the other hand, and indirectly, patients with obesity are characterized by an increase in low-grade chronic inflammation associated with mitochondrial dysfunction, an increase in oxidative stress and the suppression of the anabolic action of insulin-like growth factor-1 (IGF-1) [94,95,96,97], therefore facilitating SO [98]. As previously mentioned, chronic inflammation and pro-inflammatory cytokines are able to activate the NLRP3 protein that was found to be upregulated in KOA [50,51,52]. For this reason, the dietary approaches (i.e., diets, supplements, etc.) described in our review appear to reduce inflammation and, consequently, improve the clinical severity of KOA and SO.

### 4.2. Findings

After referral, an initial evaluation by the orthopedic specialist will confirm/reject a diagnosis of KOA as well as SO, then the patient can be assigned to one of the three different nutritional tracks based on the clinical severity:(1)Prevention Program

This track is usually reserved for patients referred to a specialized orthopedic setting, with patients reporting knee discomfort; however, after clinical imaging (i.e., X-ray), no KOA is confirmed and they are eventually classified as “Grade 0” with no further screening regarding SO required. These patients are encouraged to maintain a healthy body weight and to adhere to a cereal-rich MD, designed to reduce the risk of the onset of KOA. Moreover, supplementation with a non-EAA oral formulation (4 g three times a day for 12 weeks) was shown to improve knee discomfort in individuals, who did not receive a KOA diagnosis (Table 2; Figure 4).

(2)Lifestyle Intervention (LI) Program (72 weeks)

This is intended for patients diagnosed with Grade 1 or 2 KOA and SO, which is consequently confirmed. These patients can follow a lifestyle intervention program composed of long-term weight management (whenever indicated) by means of a LCD (i.e., 1200 kcal for females and 1500 kcal in males) with the aim of achieving WL (5%–10% over a six-month weight-loss phase), followed by a one-year weight-loss maintenance phase, as well as a supplementation of 10 g/day of EAAs for the entire duration of the WL phase (six months) and for one additional month during the weight maintenance phase. Moreover, according to the available data, a supplementation of vitamin D (less than 2000 IU/day) seems to be safe and has a beneficial effect on both KOA and SO (Table 2; Figure 4).

(3)Intensive Lifestyle Intervention (ILI) Program (Five Weeks)

Finally, this program would be appropriate for individuals diagnosed with Grade 3 or 4 KOA and the confirmation of SO. Usually, these patients are on a waiting list for TKA, based on the data as well as our clinical experience; consequently, a more aggressive bridge nutritional intervention of five weeks (three before and two after surgery) is needed. During this phase, more significant WL is required and over a shorter time; for this reason, more calorie-restricted diets are needed (i.e., VLCD or VLCKD, nearly 800 kcal/day) under a medical supervision, which have been shown to be safe and effective, as reported by recent nutritional guidelines. Similarly, a more aggressive supplementation of 20 g/day EAA over three weeks is administered (one before and two after surgery), after which the patients can switch to the “lifestyle track” (Table 2; Figure 4).

### 4.3. Clinical Implications

The findings of our narrative review may have a certain clinical implication. Firstly, in an orthopedic setting, clinicians should be aware of the importance of screening for SO in patients who receive a KOA diagnosis of. Secondly, the implementation of specific nutritional strategies, based on the clinical severity of these patients, seems to improve the clinical outcomes of KOA and SO and this should be openly discussed with patients.

### 4.4. Strengths and Limitations

To the best of our knowledge, the main strength of this narrative review is that it is the first to exclusively consider a specific population of individuals with KOA and SO in a specialized orthopedic setting and to propose a nutritional approach that can improve their clinical severity. However, our findings should also be interpreted with caution due to certain limitations. Firstly, our review is a narrative and is not systematic or meta-analytic, which could be considered a limitation, but only to a certain extent since we still believe that it is premature to conduct a systematic review on SO in this specific population due to the paucity of investigations.

### 4.5. New Directions for Future Research

To this end, there is an urgent need to direct future investigations toward new nutritional strategies that can improve the clinical outcomes related to KOA and SO in this specific population. For instance, some evidence suggests that oral hyaluronic acid (HA) supplementation increases HA concentration in the synovial fluid, reduces the inflammatory cytokines and reduces pain scores in patients with KOA and obesity [99]. However, these findings derive from very small-sampled studies [100,101]. On the other hand, recently HA was proposed to play a role against sarcopenia. In fact, a study in vitro evaluated the proliferation and hyperplasia of myocytes treated with HA and showed that the latter appeared to promote myocyte hyperplasia and proliferation as well as showed an effect against muscle atrophy, muscle damage and cell death in the sarcopenia model [102]. For this reason, future large-scale clinical trials are necessary to understand whether HA supplements can have a beneficial clinical effect on KOA and SO.

## 5. Conclusions

Sarcopenic obesity is an associated condition of KOA [20]; however, and despite this fact, it has received little attention. For this reason, and based on our findings, the baseline screening/assessment of SO should be considered in this population (i.e., KOA). Once identified, its existence should be openly discussed with patients and, based on the clinical severity of KOA, a tailored and personalized nutritional program should be proposed which can ameliorate SO.

## Figures and Tables

**Figure 1 nutrients-15-03085-f001:**
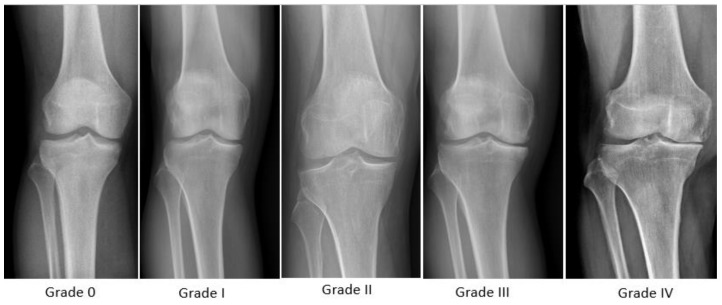
X-rays of different grades of KOA according to Kellgren and Lawrence classification system.

**Figure 2 nutrients-15-03085-f002:**
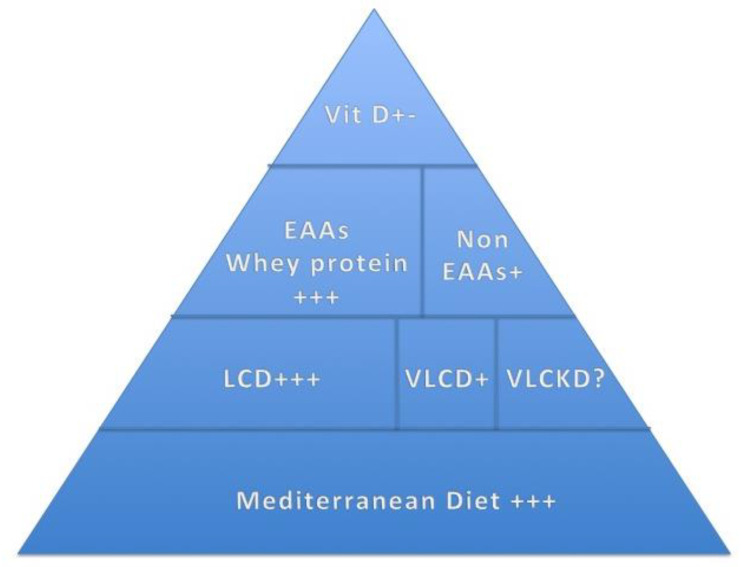
The pyramid of the nutritional management of KOA and SO. Abbreviations: Vit D = vitamin D supplementation; EAA = essential amino acid supplementation; non-EAA = non-essential amino acid supplementation; LCD = low-calorie diet; VLCD = very-low-calorie diet; VLCKD = very-low-calorie ketogenic diet. +++ = strong evidence; + = good evidence which needs replication; ? = evidence to confirm in humans; +- = contrasting evidence needs confirmation. In first place, for a successful nutritional management of KOA and SO, is the adherence to a Mediterranean diet and the achievement of weight loss whenever indicated. In second place, the supplementations with certain micronutrients appear to be beneficial (i.e., whey protein; essential and non-essential amino acids; vitamin D).

**Figure 3 nutrients-15-03085-f003:**
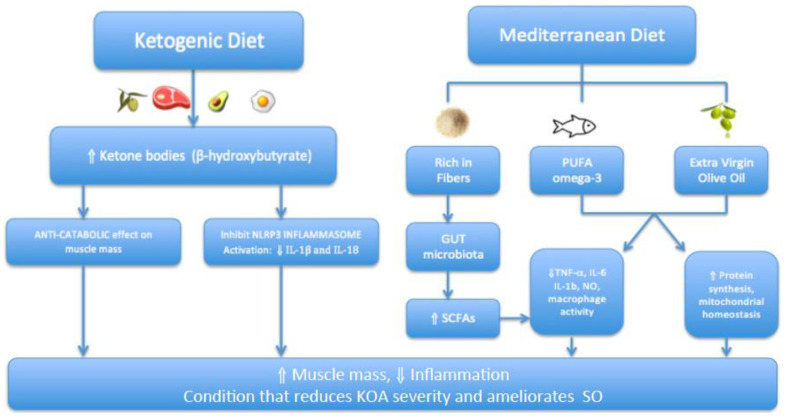
Potential mechanisms of action of the Ketogenic and Mediterranean diets in reducing KOA severity and ameliorating SO. Abbreviations: KOA = knee osteoarthritis; SO = sarcopenic obesity; SCFAs = short-chain fatty acids; NO = nitric oxide; TNF-α = tumor necrosis factor α.

**Figure 4 nutrients-15-03085-f004:**
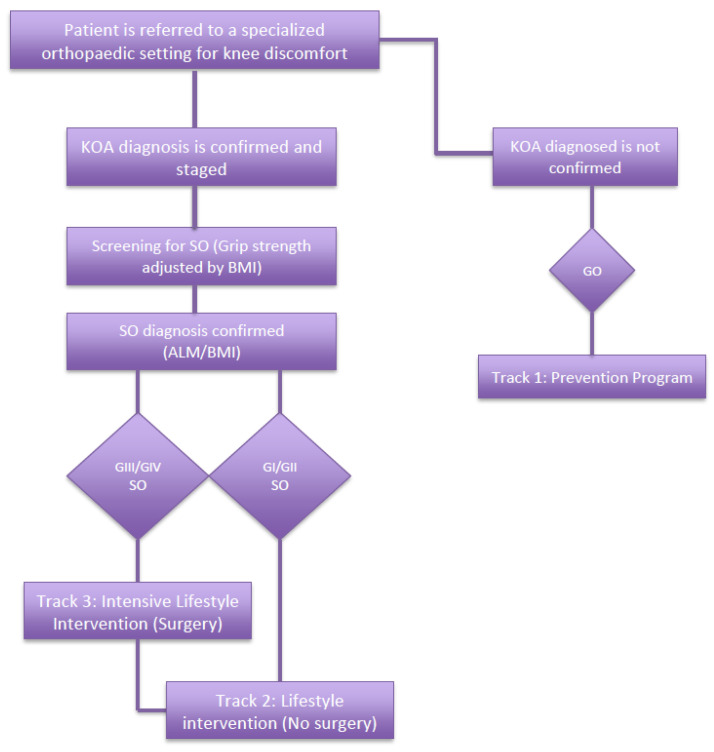
Algorithm for the nutritional approach for management of KOA.

**Table 1 nutrients-15-03085-t001:** Screening and diagnostic criteria of SO in individuals with KOA classification system.

Sarcopenic Obesity in Individuals with KOA				
		Tool	Cut-Off	
			Females	Males
Screening		Grip strength/BMI	0.65	1.1
Diagnosis		ALM/BMI	0.512	0.789
**Kellgren and Lawrence KOA Classification System**				
Grade 0	No radiological findings of OA			
Grade I	Doubtful joint space narrowing and possible osteophytic lipping			
Grade II	Certain osteophytes and possible joint space narrowing			
Grade III	Moderate multiple osteophytes, certain narrowing of joint space, some sclerosis and possible deformity of bone ends			
Grade IV	Large osteophytes, marked narrowing of joint space, severe sclerosis, and certain deformity of bone ends			

Abbreviations: KOA = knee osteoarthritis; OA = osteoarthritis; SO = sarcopenic obesity; BMI = body mass index; ALM = appendicular lean mass.

## Data Availability

Not applicable.

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
