# Peer review of "Personalized Nutritional Strategies to Reduce Knee Osteoarthritis Severity and Ameliorate Sarcopenic Obesity Indices: A Practical Guide in an Orthopedic Setting"

_nutrients, 2023, doi:10.3390/nu15143085_

Round 1
Reviewer 1 Report
Dear Editor,
I reviewd a review entitled “Common Nutritional Strategies to Reduce Knee Osteoarthritis 2
Severity and Ameliorate Sarcopenic Obesity Indices: A practical Guide in an Orthopedic Setting by
Hassan Zmerly et al
the review, based on nutritional strategies to reduce OA, is well written. however I feel there are some things to improve
major points:
-
section 3,3 2) entitled "low calorie diet" should be changed, The content does not reflect the title, the authors report the effects of weight loss and not of low calorie diet.
-
In section 3,4 3)antioxidant supplements, This section is very small, should be enriched and improved with the help of several published data
Minor points
the figure 1 should be correct, are not see images grade III and IV
Author Response
The review, based on nutritional strategies to reduce OA, is well written. However I feel there are some things to improve.
Major points:
1. Section 3,3 2) entitled "low calorie diet" should be changed. The content does not reflect the title, the authors report the effects of weight loss and not of low calorie diet. Response: We thank the reviewer. Now we changed the title of this section as suggested (Page 5, paragraph 3).
2. In section 3,4 3) Antioxidant supplements, This section is very small, should be enriched and improved with the help of several published data. Response: Despite the fact that antioxidant supplements appears not be beneficial in KAO and SO, but based on the reviewer suggestion, we developed this section (Page 7, paragraph 6).
Minor points
The figure 1 should be correct, are not see images grade III and IV. Response: Corrected as suggested (Page 3, Figure 1).
Reviewer 2 Report
The authors wrote a paper about recent nutritional strategies which can generally improve KOA clinical severity and, at the same time, ameliorate SO indices.
Although the authors write that this is a practical guide in an orthopedic setting, it is necessary to include the mechanisms of action of a particular diet, especially Mediterranean. This would increase the quality of the work, as it is now pure facts without deeper explanations, which makes the work too simple and superficial, and thus not sufficiently suitable for publication in Nutrients.
In addition, it would be necessary to include in a subtitle an explanation of the relationship between KOA and SO.
I also propose to include some schemes of specific diets with their mechanisms of action on the musculoskeletal system and metabolism.
There is something wrong with Figure 1 (cut off?)
However, this figure is not so important here, since the topic includes nutritional aspects and not diagnostic aspects. Figure 2 is also too simple, with no particular explanation.
Author Response
The authors wrote a paper about recent nutritional strategies which can generally improve KOA clinical severity and, at the same time, ameliorate SO indices.
Although the authors write that this is a practical guide in an orthopedic setting, it is necessary to include the mechanisms of action of a particular diet, especially Mediterranean. This would increase the quality of the work, as it is now pure facts without deeper explanations, which makes the work too simple and superficial, and thus not sufficiently suitable for publication in Nutrients. Response: We thank the reviewer for the valuable comment. Now we described the mechanism of action of the Mediterranean diet in details (Page 4, paragraphs 3, 4 and 5) and added wide suitable references as suggested.
In addition, it would be necessary to include in a subtitle an explanation of the relationship between KOA and SO. Response: We thank the reviewer for the valuable comment, as suggested, in the discussion section there is a subsection entitled “4.1. Interaction between KOA and SO” (Page 10, paragraph 3).
I also propose to include some schemes of specific diets with their mechanisms of action on the musculoskeletal system and metabolism. Response: We thank the reviewer for the valuable comment. Now we described the potential mechanism of action of nutrients in Mediterranean diet on musculoskeletal system (Page 5, paragraph 2).
There is something wrong with Figure 1 (cut off?). However, this figure is not so important here, since the topic includes nutritional aspects and not diagnostic aspects. Response: The screening and diagnostic tools are a part of this narrative review for both KOA and SO, and to this aim specific sections have been already mention. Moreover since the diagnostic staging system (i.e. G0, GI, GII, GIII and GIV) is a determinate to choose the suitable nutritional approach, and since this guide is addressed to the orthopedic setting, we thought that it was useful to include the X-Ray imaging of different grading of KOA according to Kellgren and Lawrence Classification System. We thank the reviewer for the comment and for understanding our point of view.
Figure 2 is also too simple, with no particular explanation. Response: More information has been added to figure 2 as suggested by the reviewer (Page 5, Figure 2).
Reviewer 3 Report
A nice and well written review.
Major comment relate to the section 3.4.1 – supplementation with vitamin D. Line 235- the information is given that supplementation with approx. 2000 IU vitamin D /day was favorable, however this statement lacks precision. In addition, the long term study performed by Australian researchers suggested that supplementation of vitamin D in a dose between 2000- 4000 IU/day for over 3 years increases the risk of fractures in older people. The authors should comment on the effects of overdosing of vitamin D on bones.
Minor comments : line 174 – the abbreviation CBT-OB is given without explanation
Several typos should be corrected : lines 147, 181, 235, 343, 348, 381, 382,
Author Response
A nice and well-written review. Response: We thank the reviewer for the appreciation.
Major comment relate to the section 3.4.1 – supplementation with vitamin D. Line 235- the information is given that supplementation with approx. 2000 IU vitamin D /day was favorable, however this statement lacks precision. Response: We thank the reviewer, now we reworded the statement to appear more clear and precise (Page 7, paragraph 3).
In addition, the long-term study performed by Australian researchers suggested that supplementation of vitamin D in a dose between 2000 - 4000 IU/day for over 3 years increases the risk of fractures in older people. The authors should comment on the effects of overdosing of vitamin D on bones. Response: We thank the reviewer for the valuable comment. Now added a paragraph (Page 7, paragraph 3) and suitable references as suggested.
Minor comments: line 174 – the abbreviation CBT-OB is given without explanation. Response: we added the explanation before the abbreviation as suggested (Page 6, paragraph 1).
Several typos should be corrected: lines 147, 181, 235, 343, 348, 381, 382. Response: Corrected as suggested.
Round 2
Reviewer 1 Report
I thank the authors for the changes they made that improved their paper
Reviewer 2 Report
I thank the authors for the changes they made to their paper. However, it still seems that the manuscript was written in a hurry. I am suggesting making one more figure suitable for a review that would include all possible mechanisms of action of the diets described, which would greatly improve the quality of the paper.
Author Response
I thank the authors for the changes they made to their paper. However, it still seems that the manuscript was written in a hurry. I am suggesting making one more figure suitable for a review that would include all possible mechanisms of action of the diets described, which would greatly improve the quality of the paper. Response: Done as suggested, now we added Figure 3, entitled “Potential mechanisms of action of the Ketogenic and Mediterranean diets in reducing KOA severity and ameliorating SO”.
Round 3
Reviewer 2 Report
The authors have addressed all queries.